# Valproic Acid Causes Redox-Regulated Post-Translational Protein Modifications That Are Dependent upon P19 Cellular Differentiation States

**DOI:** 10.3390/antiox13050560

**Published:** 2024-05-01

**Authors:** Ted B. Piorczynski, Jouber Calixto, Haley C. Henry, Kelli England, Susannah Cowley, Jackson M. Hansen, Jonathon T. Hill, Jason M. Hansen

**Affiliations:** Department of Cell Biology and Physiology, Brigham Young University, Provo, UT 84602, USA; ted.piorczynski@gmail.com (T.B.P.); jcs63@student.byu.edu (J.C.); haleys.hotmail07@gmail.com (H.C.H.); kee99@student.byu.edu (K.E.); scowley7@student.byu.edu (S.C.); jacksonmhansen8@gmail.com (J.M.H.); jhill@byu.edu (J.T.H.)

**Keywords:** valproic acid (VPA), post-translational modifications, NRF2, D3T

## Abstract

Valproic acid (VPA) is a common anti-epileptic drug and known neurodevelopmental toxicant. Although the exact mechanism of VPA toxicity remains unknown, recent findings show that VPA disrupts redox signaling in undifferentiated cells but has little effect on fully differentiated neurons. Redox imbalances often alter oxidative post-translational protein modifications and could affect embryogenesis if developmentally critical proteins are targeted. We hypothesize that VPA causes redox-sensitive post-translational protein modifications that are dependent upon cellular differentiation states. Undifferentiated P19 cells and P19-derived neurons were treated with VPA alone or pretreated with D3T, an inducer of the nuclear factor erythroid 2-related factor 2 (NRF2) antioxidant pathway, prior to VPA exposure. Undifferentiated cells treated with VPA alone exhibited an oxidized glutathione redox couple and increased overall protein oxidation, whereas differentiated neurons were protected from protein oxidation via increased *S*-glutathionylation. Pretreatment with D3T prevented the effects of VPA exposure in undifferentiated cells. Taken together, our findings support redox-sensitive post-translational protein alterations in undifferentiated cells as a mechanism of VPA-induced developmental toxicity and propose NRF2 activation as a means to preserve proper neurogenesis.

## 1. Introduction

Valproic acid (VPA) is an effective anti-epileptic drug commonly used to treat both simple and complex absence seizures [1]. Prenatal exposure to VPA increases the risk of developing fetal valproate syndrome (FVS), a condition characterized by unfavorable neurodevelopmental outcomes including decreased cognitive function and neural tube defects [2,3,4]. Despite the known teratological risks, VPA prescription rates have remained constant for females of childbearing age due to its efficacy and numerous off-label uses [5,6,7]. Strategies that protect the developing fetus while allowing the mother to continue VPA therapy have not yet been developed.

Valproic acid is a potent histone deacetylase inhibitor that causes increased acetylation across various protein targets [8,9,10,11]. One such target is superoxide dismutase 2 (SOD2)—a mitochondrial enzyme that regulates reactive oxygen species (ROS) generation. Recent findings demonstrate that VPA exposure increases SOD2 acetylation which deactivates the enzyme, resulting in an overproduction of ROS and disrupted reduction-oxidation (redox) signaling [12]. Redox dysregulation following VPA exposure is well documented in preclinical studies [13,14,15] as well as patients [16,17] and has been implicated in negative developmental outcomes [18,19,20,21]. Further research revealed that VPA-induced redox dysregulation inhibits neurogenesis in undifferentiated cells but has little effect on fully differentiated neurons, suggesting that VPA’s toxicity may be dependent upon cellular differentiation states [22]. Pretreatment of both cells and embryos with 3H-1,2-dithiole-3-thione (D3T), an inducer of the nuclear factor erythroid 2-related factor 2 (NRF2) antioxidant pathway, preserves redox homeostasis and promotes proper development, further supporting the theory of VPA-induced teratogenicity through disrupted redox signaling [22,23]. While recent findings certainly implicate redox dysregulation in negative developmental outcomes, the exact oxidative mechanisms responsible for FVS remain to be elucidated.

Proper development requires tightly controlled oxidative conditions as redox signals regulate critical cellular processes including proliferation, differentiation, and apoptosis [24,25,26]. Redox signaling commonly occurs via oxidative post-translational modifications (oxPTMs) that act as activation or inactivation “switches” of protein activity [27]. The maintenance of specific oxPTMs through redox control is essential to preserve proper protein function and support normal embryogenesis [28]. With previous findings confirming that undifferentiated cells are especially susceptible to VPA-induced redox dysregulation [22], we hypothesize that VPA causes redox-sensitive post-translational protein modifications that are dependent upon cellular differentiation states. Additionally, we propose that the induction of the NRF2 antioxidant pathway through D3T pretreatment preserves redox homeostasis and consequently reduces oxPTMs caused by VPA exposure.

Changes in redox potentials and oxPTM levels were assessed using high-performance liquid chromatography and redox immunoblotting techniques, respectively, in treated undifferentiated P19 cells and P19-derived neurons. Undifferentiated cells treated with VPA exhibited an oxidized glutathione redox couple and increased overall protein oxidation. In contrast, VPA treatment had no effect on the glutathione redox couple or protein oxidation in differentiated neurons but resulted in an increase in protective protein *S*-glutathionylation. Pretreatment with D3T prevented VPA-induced protein oxidation and decreased transcriptional dysregulation in undifferentiated cells as measured via RNA sequencing. Our findings propose redox-sensitive post-translational protein alterations as a potential mechanism of FVS and demonstrate the effectiveness of NRF2-mediated redox regulation during early neurogenesis.

## 2. Materials and Methods

### 2.1. Cell Culture

P19 mouse embryonal carcinoma cells were purchased from the American Type Culture Collection and cultured in growth medium comprised of Alpha Minimum Essential Medium (Genesee Scientific, El Cajon, CA, USA) supplemented with bovine calf serum (BCS; 7.5% *v*/*v*; American Type Culture Collection, Manassas, VA, USA), fetal bovine serum (FBS; 2.5% *v*/*v*; Genesee Scientific), penicillin (100 U/mL; Genesee Scientific), and streptomycin (100 μg/mL; Genesee Scientific). Cells were grown in a 37 °C humidified incubator maintained at 5% CO_2_.

### 2.2. Neuronal Differentiation

P19 cells were differentiated into neurons over a four-day-long process as described previously [29]. In summary, cells were seeded at a density of 1 × 10^6^ cells per dish in a differentiation medium comprised of Dulbecco’s Modified Eagle’s Medium (DMEM; Genesee Scientific) supplemented with FBS (5% *v*/*v*), antibiotics, and retinoic acid (1 μM; Sigma-Aldrich, St. Louis, MO, USA) to promote neurogenesis. Cells were cultured on a rotating stage to promote aggregate generation [30]. The medium was changed on day two of differentiation. On day four, aggregates were plated on cell culture plates in a post-differentiation maintenance medium comprised of DMEM supplemented with FBS (10% *v*/*v*) and antibiotics. Differentiated cells were cultured for two additional days before being used for subsequent experiments.

### 2.3. Cell Treatment

Treatment with 10 μM D3T (LKT Labs, St. Paul, MN, USA) or 5 mM VPA (Sigma-Aldrich) had no effect on cell viability (Appendix A) but increased markers of NRF2 induction (Appendix A) and ROS production (Appendix A), respectively. For all subsequent experiments, both undifferentiated P19 cells and P19-derived neurons were pretreated with D3T (10 μM) or vehicle (dimethyl sulfoxide, DMSO; Sigma-Aldrich) for 12 h, exposed to VPA (5 mM) or vehicle (medium) for 6 h, and either imaged or collected for subsequent analyses. Separate cells were treated with hydrogen peroxide (H_2_O_2_; 200 μM; Sigma-Aldrich) for 10 min as a positive control.

### 2.4. Redox Couple Chromatography

In order to evaluate changes in the intracellular redox environment, treated cells were collected in perchloric acid (5% *v*/*v*; Sigma-Aldrich) and boric acid (0.2 M; Sigma-Aldrich) containing γ-glutamylglutamate (10 µM; Sigma-Aldrich). Intracellular concentrations of cysteine (Cys), cystine (CySS), glutathione (GSH), and glutathione disulfide (GSSG) were measured by reverse-phase, high-performance liquid chromatography as *S*-carboxymethyl, *N*-dansyl derivatives normalized to γ-glutamylglutamate [31]. Proteins were acid-precipitated, and samples were centrifuged at 16,000× *g* for 5 min, after which the soluble fraction containing free Cys, CySS, GSH, and GSSG was derivatized with dansyl chloride (Sigma-Aldrich). Samples were separated using an e2695 Separations Module (Waters, Milford, MA, USA) fitted with a Supelcosil LC-NH2 5 µm column (Sigma-Aldrich) and peak detection was determined using a 2474 FLR Detector (Waters). The Cys/CySS and GSH/GSSG redox potentials (E_h_) were calculated by the Nernst equation using Cys, CySS, GSH, and GSSG intracellular concentrations [32].

### 2.5. RNA-Sequencing

Treated cells were collected in TRIzol Reagent (Thermo Fisher Scientific, Waltham, MA, USA) and RNA was extracted using Direct-zol RNA Miniprep Kits (Zymo Research, Irvine, CA, USA). Both RNA concentration and integrity were measured using a NanoDrop ND-1000 Spectrophotometer (Thermo Fisher Scientific). All samples submitted for sequencing had an RNA integrity number > 9. Library construction and sequencing were performed by NovoGene (Sacramento, CA, USA); poly(A)-containing RNA was isolated from 500 ng of total RNA, fragmented, and synthesized into cDNA. Unique indexing adapters were ligated to the cDNA, and the samples were expanded by PCR amplification. Each library was sequenced as paired-end, 150-base-pair reads using a NovaSeq 6000 System (Illumina, San Diego, CA, USA).

### 2.6. Differential Gene Expression Analysis and Clustering

Read quality was assessed using FastQC [33] (v0.12), and the reads from each sample were mapped independently to the mouse reference transcriptome (Ensembl, Mus musculus version 103) using Kallisto [34] (v0.48.0). The results of transcript quantification were summarized to genes using Tximeta [35] (v1.16.1). Differential gene expression was evaluated using the Wald test through DESeq2 [36] (v1.40.2). Genes with an adjusted *p*-value < 0.05 were designated as differentially expressed. Clust [37] (v1.18.0) was used to analyze co-regulated genes, and an enrichment analysis of select clusters was performed using ClusterProfiler [38] (v4.8.2). Data were plotted using ggplot2 [39] (v3.5.0).

### 2.7. Redox Immunoblotting

Three biotinylated probes were used to evaluate oxPTMs: (1). biotinylated iodoacetamide (BIAM; Invitrogen, Carlsbad, CA, USA) was used to measure reduced protein thiols; (2). 3-(2,4- dioxo cyclohexyl)propyl biotin (DCP-Bio1; Kerafast, Boston, MA, USA) was used for protein sulfenic acid detection; and (3). biotinylated glutathione ethyl ester (BioGEE; Nanosoft Biotechnology, Coventry, RI, USA) was employed to analyze *S*-glutathionylation. Treated cells were collected in RIPA lysis buffer (Sigma-Aldrich) containing protease inhibitors (Roche Diagnostics, Indianapolis, IN, USA) and additional components depending on the particular oxPTM being analyzed. To measure reduced protein thiols, cells were collected in lysis buffer containing BIAM (250 μM), incubated in a 37 °C water bath for 20 min, treated with iodoacetamide (IAM; 5 mM; Sigma-Aldrich), and incubated at room temperature for 10 min. To measure sulfenic acid formation, cells were collected in lysis buffer containing IAM (5 mM) and DCP-Bio1 (500 μM) then incubated on ice for 1 h. To measure *S*-glutathionylation, cells were preloaded with BioGEE (250 μM) for 2 h prior to VPA exposure and then collected in lysis buffer containing IAM (5 mM). Protein concentrations were determined using BCA Protein Assay Kits (Thermo Fisher Scientific). Equal amounts of protein from all samples were separated by non-reducing polyacrylamide gel electrophoresis and each sample was run in duplicate: one sample set was stained with GelCode Blue Stain Reagent (Thermo Fisher Scientific) to visualize total protein loading while the other set was transferred onto a nitrocellulose membrane (Genesee Scientific). Membranes were probed with a streptavidin Alexa Fluor 680 conjugate (Thermo Fisher Scientific #S21378) diluted 1:5000 in phosphate-buffered saline (PBS) with Tween 20 (PBST; 0.1% *v*/*v*; Sigma-Aldrich) at room temperature for 1.5 h. Membranes were washed with PBST three consecutive times and then imaged on an Odyssey CLx Imaging System (LI-COR Biosciences, Lincoln, NE, USA) and quantified using Image Studio (v5.5, LI-COR Biosciences). In order to account for the background signal, blank control samples that were not incubated with any oxPTM probes were run on each gel and their normalized signal intensities were subtracted from all sample signal intensities run on the same blot. The oxPTM expression of each sample was normalized to its respective GelCode-stained load control before comparison.

### 2.8. Confocal Microscopy

To visualize protein sulfenic acid formation, cells were washed with PBS and incubated with rhodamine B [4-[3-(2,4-dioxocyclohexyl)propyl]carbamate] piperazine amide (DCP-Rho1; 10 μM; Kerafast) in a 37 °C humidified incubator protected from light for 10 min. Cells were washed with PBS three times and then suspended in a growth medium containing HEPES (25 mM; Thermo Fisher Scientific) but without phenol red. To visualize *S*-glutathionylation, cells were first preloaded with BioGEE (250 μM) for 2 h prior to VPA exposure. After treatment, cells were washed with PBS, fixed in paraformaldehyde (4%; Sigma-Aldrich) at room temperature for 10 min, permeabilized with Triton X-100 (0.5%; Sigma-Aldrich) for 10 min, and probed with streptavidin Alexa Fluor 488 conjugate (Thermo Fisher Scientific # S11223) at room temperature for 30 min. Both DCP-Rho1- and BioGEE-treated cells were incubated with Hoeschst dye (10 ug/mL; Thermo Fisher Scientific) for 10 min, washed three additional times, and imaged on a FV1000 confocal microscope (Olympus, Center Valley, PA, USA).

### 2.9. Statistical Analyses

The R computational software environment (v4.3.3) was used for all statistical analyses. Homoscedasticity for each experiment was assessed using a Shapiro–Wilk test. Statistical comparisons were performed using a one-way analysis of variance (ANOVA) followed by pairwise t-tests using the Bonferroni correction. Quantitative data are presented as means ± standard error of the mean (SEM). Asterisks denote a statistically significant difference (* = *p* < 0.05, ** = *p* < 0.01, and *** = *p* < 0.001).

### 2.10. Graphic Design

The graphical abstract, Figure 1a, and Figure 6a,b were constructed using BioRender (https://www.biorender.com, accessed on 9 September 2023). All graphs and plots were generated in R and compiled, alongside confocal and immunoblot images, in Inkscape (v1.2.2).

## 3. Results

### 3.1. Valproic Acid (VPA) Alters the Glutathione Redox Couple in Undifferentiated Cells

P19 mouse embryonal cells can be differentiated into neurons (Appendix A) and are commonly used as a model of neurogenesis [29]. To assess treatment effects on different differentiation states, undifferentiated P19 cells and P19-derived neurons were pretreated with D3T or vehicle, exposed to VPA, and collected for subsequent analyses (Figure 1a). Many redox potentials work together to maintain the intracellular redox environment; GSH/GSSG and Cys/CySS are two of the most abundant redox couples and are commonly measured to assess the general intracellular redox environment [25]. Undifferentiated cells treated with VPA exhibited unchanged GSH concentrations (Figure 1b), increased GSSG concentrations (Figure 1d), and a more oxidizing GSH/GSSG E_h_ (Figure 1f) compared to vehicle-treated controls. Activation of the NRF2 antioxidant pathway through D3T pretreatment prevented all VPA-induced oxidative changes. Differentiated neurons treated with VPA displayed no significant changes in the GSH/GSSG redox couple. Additionally, VPA exposure resulted in no significant changes in Cys concentrations (Figure 1c), CySS levels (Figure 1e), or Cys/CySS E_h_ (Figure 1g) in undifferentiated or differentiated cells. In summary, VPA causes oxidation of the GSH/GSSG redox couple that is ameliorated with D3T pretreatment in undifferentiated cells but has no effect on the Cys/CySS redox couple regardless of the cellular differentiate state.

### 3.2. Valproic Acid Increases Protein Oxidation in Undifferentiated Cells but Not in Differentiated Neurons

Valproic acid increases ROS generation in P19 cells (Appendix A) but fails to induce the NRF2 antioxidant pathway [40,41]. Fortunately, D3T upregulates various antioxidant genes (Appendix A) through its potent NRF2 induction (Appendix A) and consequently decreases VPA-induced ROS production (Appendix A). Uncontrolled ROS production can cause oxPTMs that result in altered protein function and disrupted embryogenesis [28]. Undifferentiated cells and differentiated neurons were assessed for overall protein oxidation using redox immunoblotting techniques. Undifferentiated cells treated with VPA exhibited significantly increased protein oxidation compared to vehicle-treated controls (Figure 2a–c). Neither VPA nor H_2_O_2_ exposure affected overall protein oxidation in P19-derived neurons (Figure 2d–f), demonstrating differentiated cells’ resilience to oxidative perturbations through potential protective mechanisms such as *S*-glutathionylation. Pretreatment of undifferentiated cells with D3T prevented protein oxidation. In conclusion, undifferentiated cells are more susceptible to VPA-induced protein oxidation compared to their differentiated counterparts.

### 3.3. Sulfenic Acid Formation Is Increased in Undifferentiated Cells but Decreased in Neurons Treated with VPA

Sulfenic acid formation on active-site cysteines often acts as a “switch” for protein activity and function [42]. Treatment with VPA significantly increased sulfenic acid formation in undifferentiated cells, as measured via confocal microscopy (Figure 3a) and redox immunoblotting (Figure 3b–d). In contrast, differentiated neurons treated with VPA displayed decreased sulfenic acid formation compared to vehicle-treated controls (Figure 3e–h). Pretreatment with D3T prevented VPA-induced modifications in both undifferentiated and differentiated cells. Thus, undifferentiated cells are susceptible to protein sulfenic acid formation following VPA exposure but are protected with prior NRF2 induction.

### 3.4. Protein S-Glutathionylation Is Unchanged in Undifferentiated Cells but Increased in Differentiated Neurons following VPA Exposure

Protein *S*-glutathionylation is an important post-translational modification that protects cysteines from over-oxidation [42]. Valproic acid treatment did not lead to differential *S*-glutathionylation expression in undifferentiated cells as measured via confocal microscopy (Figure 4a) or redox immunoblotting (Figure 4b–d) but did increase *S*-glutathionylation in differentiated neurons (Figure 4e–h).

Activation of the NRF2 pathway prevented VPA-induced *S*-glutathionylation in neurons. Increased *S*-glutathionylation in differentiated neurons may therefore aid in preventing protein oxidation following VPA exposure.

### 3.5. Nuclear Factor Erythroid 2-Related Factor 2 (NRF2) Activation Protects Neurodevelopmental Transcription Pathways from VPA Exposure in Undifferentiated Cells

Changes in cellular redox states and oxPTMS can affect transcriptional regulatory mechanisms [43]. Differential gene expression between treatment conditions was assessed via RNA sequencing in both undifferentiated and differentiated samples (Appendix A). Sequencing identified over 3500 differentially expressed genes comparing VPA alone to D3T and VPA combination treatment in undifferentiated cells (Figure 5a) but found only one differential gene comparing the same treatments in differentiated neurons (Figure 5b). Similar transcriptional patterns between treatment groups were observed using heatmaps for undifferentiated (Figure 5c) and differentiated samples (Figure 5d). Clustering analyses were used to identify genes that were similarly affected by VPA exposure and protected with D3T pretreatment. Undifferentiated gene clusters 3, 4, 5, 7, 10, 11, and 12 show transcriptional correction with D3T and VPA combination treatment compared to VPA alone (Figure 5e). No differentiated gene clusters displayed significant correction from VPA exposure with D3T pretreatment (Appendix A). The genes from the aforementioned undifferentiated clusters were combined and assessed using a gene ontology analysis (Figure 5f) to reveal numerous neurodevelopmental pathways that are transcriptionally protected through NRF2 induction.

## 4. Discussion

Recent findings support redox re-regulation as an approach to protect VPA-treated embryos from congenital malformations [23], yet the specific biological pathways responsible for conferring developmental protection remain unknown. Redox signaling frequently occurs via thiol/disulfide redox couples that play critical roles in cell signaling, protein regulation, and macromolecular structure [44]. Each redox couple has a corresponding steady-state E_h_ and shifts in these equilibrium states result in the rapid and dynamic regulation of precise cellular processes [45,46]. For example, the independent regulation of specific redox couples is observed during cellular differentiation; differentiation of colon epithelial cells correlates with an oxidation of the GSH/GSSG E_h_, but the thioredoxin 1 reduced/oxidized E_h_, another common redox couple, is unaffected [47]. Similarly, human mesenchymal stem-cell redox ontogenies have been shown to correlate with specific differentiation fates; osteogenesis is associated with a reduction in the Cys/CySS E_h_, whereas adipogenesis requires a mild oxidation [48]. Though the GSH/GSSG E_h_ ultimately becomes oxidized to a similar extent during both adipogenesis and osteogenesis, the rate by which the redox potential changes is vastly different—it is rapidly oxidized during adipogenesis but shifts much more gradually during osteogenesis. These findings support the understanding that redox couples are independently regulated and that each couple dynamically regulates specific targets of a singular cellular process to yield an explicit outcome.

In the present study, VPA treatment in undifferentiated P19 cells oxidized the GSH/GSSG E_h_ but had no effect on the Cys/CySS E_h_. Interestingly, VPA treatment had no effect on either the GSH/GSSG E_h_ or Cys/CySS E_h_ in differentiated neurons, implying that the GSH/GSSG redox couple may be the primary redox node that controls neurogenesis in undifferentiated cells. Undifferentiated P19 cells may be especially susceptible to oxidative insults due to their inherently more reducing basal GSH/GSSG E_h_ compared to their differentiated counterparts. The GSH/GSSG E_h_ undergoes a significant (> +15 mV) change over the course of P19 neurogenesis and is carefully regulated to maintain proper differentiation; P19 neurogenesis is disrupted by VPA-induced redox disruption but is protected with GSH/GSSG E_h_ stabilization through NRF2 induction [22]. Other work from our lab using redox-sensitive probes to analyze GSH/GSSG redox states confirmed differential oxidation and redox rebounding profiles in undifferentiated and differentiated cells following oxidant exposure [26]. The observed pattern is similar to that described during embryonic development, where the expression of antioxidant enzymes and redox-regulating activities are upregulated as development advances and are likely a means to prepare the embryo for extrauterine life [49].

Shifts in redox couple steady states can affect macromolecular trafficking, structure, and function [50]. Proteins contain two common functional groups—Cys and methionine—which undergo reversible redox reactions. While the oxidation of methionine can occur in association with redox signaling, research has predominantly focused on the analysis of Cys as the primary redox-sensitive element in proteins [27]. Changes in redox couple steady states may lead to the oxidative modification of certain protein thiols (-SH) that control macromolecular function [52,53]. To assess differential protein thiol modifications, undifferentiated P19 cells and P19-derived neurons were treated with VPA and analyzed for three different types of oxPTMs: reduced Cys thiols, sulfenic acid formation, and protein *S*-glutathionylation. Under oxidative conditions, Cys thiols can be progressively oxidized into sulfenic (-SOH), sulfinic (-SO_2_H), and sulfonic acids (-SO_3_H). Protein sulfenic acids are effectively managed through oxidoreductases that quickly convert them back to fully reduced thiols [53]. Conversely, sulfinic acids are only slowly reversible while sulfonic acids are irreversible and lead to protein degradation [54]. To protect sulfenic acids from over-oxidation into sulfinic or sulfonic acids, oxidized protein thiols react spontaneously with GSH to form protective *S*-glutathione derivatives [55]. Thiols remain *S*-glutathionylated until GSH is enzymatically removed at a later time, providing timely protection from irreversible oxidation and premature protein degradation. Results here show that undifferentiated P19 cells treated with VPA exhibited decreased protein *S*-glutathionylation that corresponded with increased sulfenic acid formation and overall protein oxidation. Different effects were observed in neurons: sulfenic acid formation decreased, *S*-glutathionylation increased, and overall protein oxidation was unchanged following VPA treatment. Taken together, we propose that undifferentiated P19 cells are more susceptible to VPA-induced protein over-oxidation due to reduced *S*-glutathionylation capabilities (Figure 6a). Differentiated P19 cells, on the other hand, exhibit increased protein *S*-glutathionylation that likely preserves protein function and supports timely protein recovery (Figure 6b).

The inducible transcription factor NRF2 is activated during periods of oxidative stress and initiates the antioxidant response through the upregulation of numerous detoxification enzymes [56]. Although VPA increases ROS generation and causes widespread redox disruption, it surprisingly fails to activate the NRF2 antioxidant pathway [40,41]. We observed that pretreatment with the synthetic NRF2 inducer, D3T, stabilizes the GSH/GSSG redox couple and all measured oxPTM levels in VPA-treated undifferentiated cells. In the context of redox regulation, both sulfenic acid formation [57] and protein S-glutathionylation [60] have been identified as important signaling modifications with numerous targets and effects. As such, the aberrant levels of oxPTMs observed following VPA exposure, which likely play a critical role in causing FVS, may be prevented through NRF2 activation and the subsequent preservation of normal, developmentally healthy redox states. Though D3T is a synthetic compound, several other NRF2 inducers are found naturally. For example, sulforaphane is extracted from cruciferous vegetables and can easily cross the blood–brain barrier to exert various neuroprotective effects through its NRF2 activation [59,60]. One advantage to using naturally found NRF2 inducers is that they can be easily incorporated into maternal diets and are inexpensive compared to potential pharmaceutical therapies. Maternal diets during pregnancy vary considerably around the world and may therefore benefit from dietary recommendations that include natural NRF2 inducers [61]. Additional work is necessary to evaluate the numerous naturally found NRF2 inducers as a way to preserve proper embryonic development.

Mechanistic studies demonstrate that VPA relieves histone deacetylase-dependent transcriptional repression and regulates the expression of various genes involved in proper neurodevelopment [62]. In the present study, VPA exposure caused considerable differential gene expression in both undifferentiated and differentiated P19 cells. Interestingly, D3T pretreatment decreased VPA-induced transcriptional dysregulation in undifferentiated cells but had little effect in differentiated neurons. Stabilization of the redox environment through NRF2 activation likely preserves appropriate oxPTMs on developmentally critical transcriptional regulators. Indeed, many transcription factors, including p53 [63], AP-1 [64], and NF-KB [65], and other transcriptional regulators are subject to post-translational redox signaling [66]. The gene-enrichment analyses presented here reveal numerous neurodevelopmental pathways protected from VPA exposure through D3T pretreatment in undifferentiated cells. Though D3T does not confer transcriptional protection in neurons, the increased protein *S*-glutathionylation observed in the differentiated phenotype may protect critical transcriptional regulators from the over-oxidation seen in their undifferentiated counterparts. Taken together, VPA causes transcriptional dysregulation that is reduced with D3T pretreatment, implicating NRF2 induction as a means to prevent negative transcriptional alterations through post-translational modifications.

A limitation of the present study is the exclusive use of P19 cells for all of our experiments, though they are regarded as an excellent model system for analyzing neural differentiation [67,68]. Furthermore, the use of a single cell line for all experiments grants deconvoluted results but does not fully capture the heterogeneous and complex cellular environment of embryogenesis. Neuronally differentiated murine P19 cells are functionally similar to primary neural cells [69]; nevertheless, our findings require validation in more relevant human models. While the present study highlights the importance of maintaining oxPTMs across general proteomic targets, it does not identify the precise proteins affected by toxicant exposure. The use of protein-identification methodologies will pinpoint the exact, redox-regulated targets responsible for FVS and NRF2-mediated protection. Although VPA has been the focus of this work, many developmental toxicants, including pharmaceutics, metals, recreational drugs, and environmental contaminants, have been shown to induce ROS production and cause redox dysregulation [70]. The identification of developmentally critical, redox-regulated proteins may therefore help improve therapies against numerous other developmental toxicants. Clearly, additional work is required to fully understand the role of redox regulation in embryogenesis.

In summary, our findings demonstrate that undifferentiated P19 cells are more susceptible to VPA-induced redox couple disruption compared to their neuronally differentiated counterparts. Consequently, VPA-treated undifferentiated cells exhibit increased levels of sulfenic acid formation and increased overall protein oxidation compared to differentiated neurons. The induction of the NRF2 antioxidant pathway prevents the observed post-translational protein modifications and protects neurodevelopmental transcription pathways. Together, these data support redox-sensitive post-translational protein alterations as a mechanism of VPA-induced developmental toxicity and propose NRF2 activation as a means to preserve proper neurogenesis.

## Figures and Tables

**Figure 1 antioxidants-13-00560-f001:**
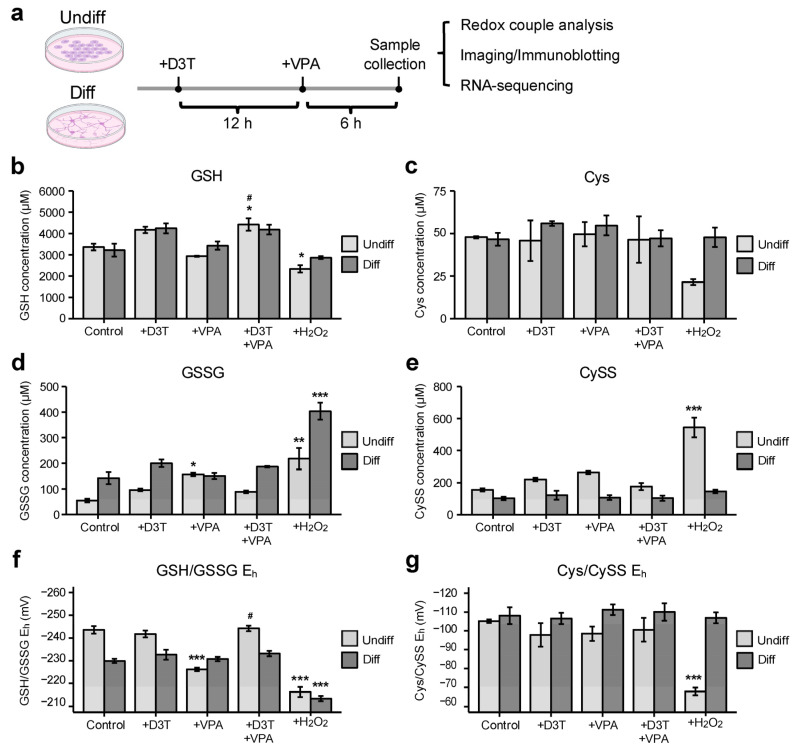
Valproic acid alters the glutathione redox couple in undifferentiated cells. (**a**) Undifferentiated P19 cells (undiff) and P19-derived neurons (diff) were pretreated with 3H-1,2-dithiole-3-thione (D3T; 10 μM) or vehicle for 12 h, exposed to valproic acid (VPA; 5 mM) or vehicle for 6 h and either imaged or collected for subsequent analyses. (**b**,**d**,**f**) Undifferentiated and differentiated cells displayed variable glutathione (GSH) concentrations (*n =* 3; (**b**)), glutathione disulfide (GSSG) levels (*n =* 3; (**d**)), and GSH/GSSG redox potentials (E_h_; *n =* 3; (**f**)) following treatment. (**c**,**e**,**g**) In contrast, neither undifferentiated nor differentiated cells exhibited significant changes in cysteine (Cys; *n =* 3; (**c**)), cystine (CySS; *n =* 3; (**e**)), or Cys/CySS E_h_ (*n =* 3; (**g**)). Brief exposure to hydrogen peroxide (H_2_O_2_) was used as a positive control (**b**–**g**). Data are presented as means ± standard error of the mean (SEM; (**b**–**g**)). Statistical comparisons were made using a one-way analysis of variance (ANOVA) followed by a pairwise t-test using the Bonferroni correction ((**b**–**g**)). Asterisks denote a statistically significant difference (* = *p* < 0.05, ** = *p* < 0.01, and *** = *p* < 0.001) compared to respective vehicle-treated controls, whereas number signs denote a statistically significant difference (# = *p* < 0.05) in combined D3T and VPA treatment compared to VPA alone ((**b**–**g**)).

**Figure 2 antioxidants-13-00560-f002:**
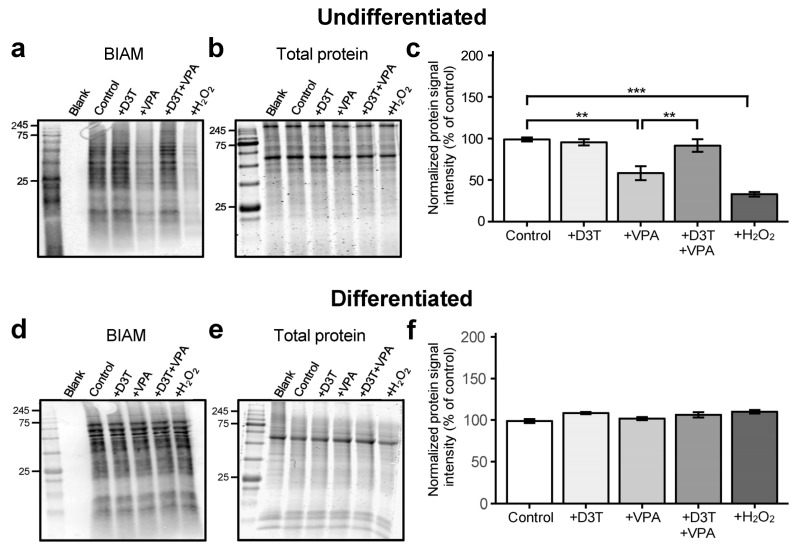
Valproic acid increases protein oxidation in undifferentiated cells but not in differentiated neurons. Undifferentiated P19 cells and P19-derived neurons were treated with D3T and VPA and then labeled with biotinylated iodoacetamide (BIAM), a marker of reduced protein thiols. (**a**–**c**) Undifferentiated cells were assessed for BIAM signal intensity (**a**), normalized to their respective GelCode Blue-stained samples run in parallel (**b**), and analyzed using the normalized protein signal intensity (*n =* 3; (**c**)). (**d**–**f**) Differentiated neurons were also assessed for BIAM signal intensity (**d**), normalized to their respective GelCode Blue-stained samples (**e**), and analyzed using the normalized protein signal intensity (*n =* 3; (**f**)). Normalized BIAM signal intensity is an inverse measure of overall protein oxidation. Brief exposure to H_2_O_2_ was used as a positive control (**a**–**f**). Data are presented as means ± SEM. (**c**,**f**). Statistical comparisons were made using a one-way ANOVA followed by a pairwise t-test using the Bonferroni correction (**c**,**f**). Asterisks denote a statistically significant difference (** = *p* < 0.01, and *** = *p* < 0.001; (**c**,**f**)).

**Figure 3 antioxidants-13-00560-f003:**
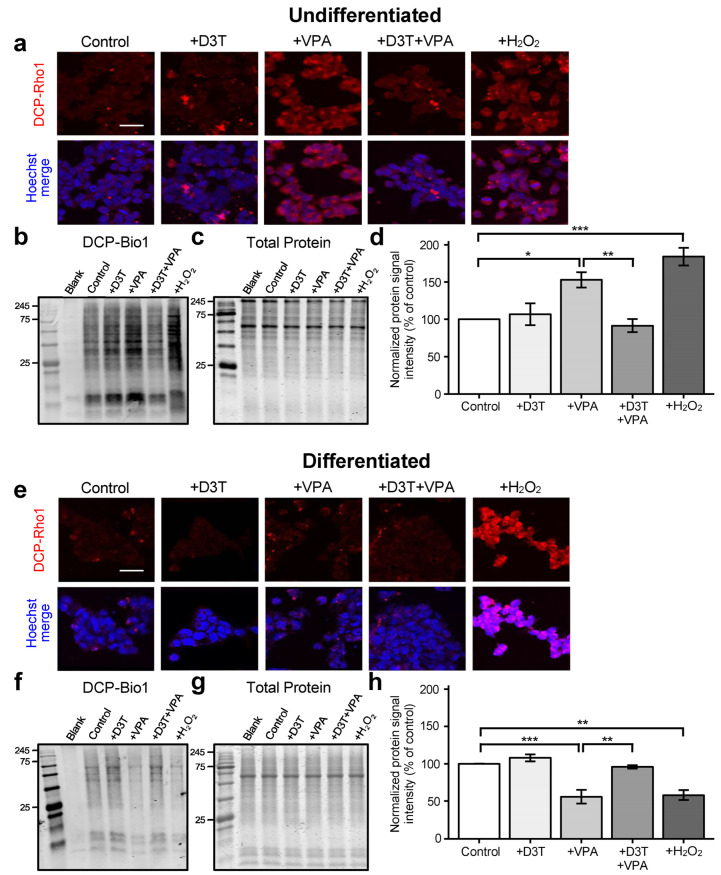
Sulfenic acid formation is increased in undifferentiated cells but decreased in neurons treated with VPA. (**a**) Undifferentiated P19 cells were treated, stained with DCP-Rho1—a probe used to detect sulfenic acid formation—and imaged using confocal microscopy. (**b**–**d**) Undifferentiated cells were further assessed for sulfenic acid formation through 3-(2,4-dioxo cyclohexyl)propyl biotin (DCP-Bio1) signal intensity (**b**), normalized to their respective GelCode Blue-stained samples run in parallel (**c**), and analyzed using the normalized protein signal intensity (*n =* 3; (**d**)). (**e**) Differentiated P19 cells were similarly stained with DCP-Rho1 and imaged. (**f**–**h**) Differentiated neurons were assessed for DCP-Bio1 signal intensity (**f**), normalized to their respective GelCode Blue-stained samples (**g**), and analyzed using the normalized protein signal intensity (*n =* 3; (**h**)). Normalized DCP-Bio1 signal intensity is a direct measure of protein sulfenic acid formation. Brief exposure to H_2_O_2_ was used as a positive control (**a**–**h**). Scale bars represent 50 μm (**a**,**e**). Images are representative of three independent experiments (**a**,**e**) and data are presented as means ± SEM (**d**,**h**). Statistical comparisons were made using a one-way ANOVA followed by a pairwise t-test using the Bonferroni correction (**d**,**h**). Asterisks denote a statistically significant difference (* = *p* < 0.05, ** = *p* < 0.01, and *** = *p* < 0.001; (**d**,**h**)).

**Figure 4 antioxidants-13-00560-f004:**
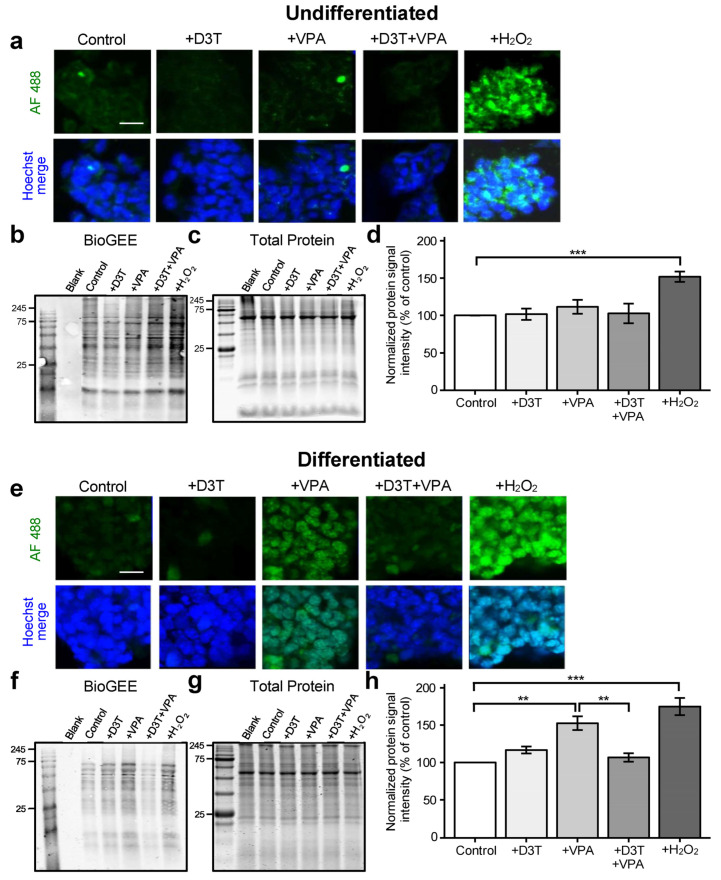
Protein *S*-glutathionylation is unchanged in undifferentiated cells but increased in differentiated neurons following VPA exposure. Undifferentiated cells and differentiated neurons were treated and collected for protein *S*-glutathionylation detection using biotinylated glutathione ethyl ester (BioGEE), a GSH analog. (**a**) Undifferentiated cells were stained with a streptavidin-conjugated fluorophore, Alexa Fluor 488 (AF 488), to detect BioGEE and then imaged using confocal microscopy. (**b**–**d**) Undifferentiated cells were assessed for BioGEE signal intensity (**b**), normalized to their respective GelCode Blue-stained samples run in parallel (**c**), and analyzed using the normalized protein signal intensity (*n =* 3; (**d**)). (**e**) Differentiated cells were similarly probed with BioGEE and imaged. (**f**–**h**) Differentiated neurons were assessed for BioGEE signal intensity (**d**), normalized to their respective GelCode Blue-stained samples (**g**), and analyzed using the normalized protein signal intensity (*n =* 3; (**h**)). Normalized BioGEE signal intensity is a direct measure of protein *S*- glutathionylation. Brief exposure to H_2_O_2_ was used as a positive control (**a**–**h**). Scale bars represent 50 μm (**a**,**e**). Images are representative of three independent experiments (**a**,**e**) and data are presented as means ± SEM (**d**,**h**). Statistical comparisons were made using a one-way ANOVA followed by a pairwise t-test using the Bonferroni correction (**d**,**h**). Asterisks denote a statistically significant difference (** = *p* < 0.01, and *** = *p* < 0.001; (**d**,**h**)).

**Figure 5 antioxidants-13-00560-f005:**
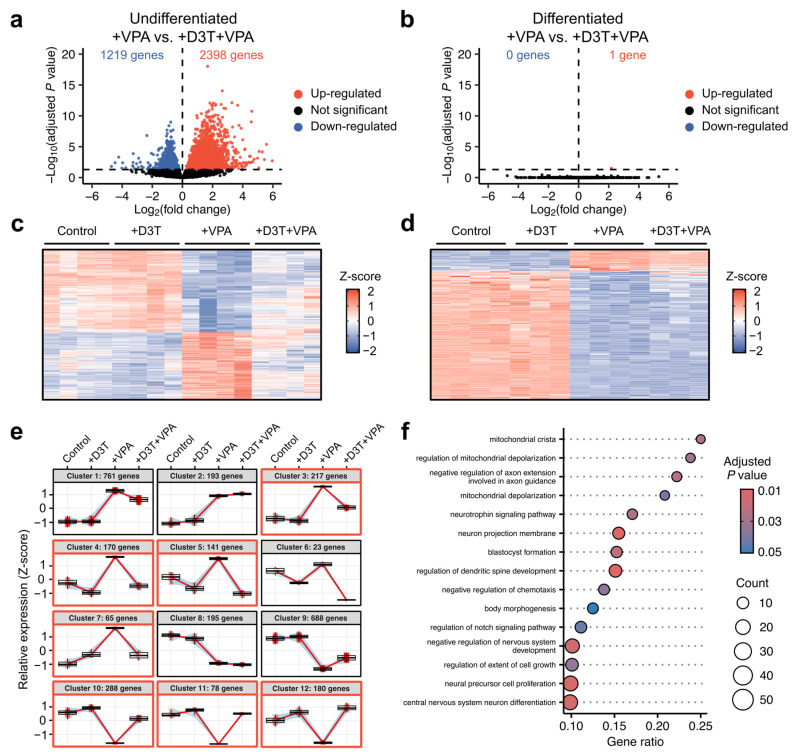
Nuclear factor erythroid 2-related factor 2 (NRF2) activation protects neurodevelopmental transcription pathways from VPA exposure in undifferentiated cells. Undifferentiated cells and differentiated neurons were treated and analyzed using bulk RNA-sequencing. Genes with adjusted *p* values < 0.05 were considered differentially expressed. (**a**) Over 3500 genes were differentially expressed when comparing combination D3T and VPA treatment to VPA alone in undifferentiated cells (*n =* 4). (**b**) Conversely, only one gene was differentially expressed when comparing the same treatment conditions in differentiated neurons (*n =* 3–4). (**c**,**d**) Heatmaps display distinct transcriptional patterns across treatment conditions in undifferentiated (**c**) and differentiated (**d**) samples. (**e**) Gene clustering across undifferentiated samples reveals numerous gene groups that are similarly affected by VPA exposure and protected with D3T pretreatment; gene clusters 3, 4, 5, 7, 10, 11, 12 (highlighted in red) show gene expression correction with combination D3T and VPA treatment compared to VPA alone. (**f**) Genes from the aforementioned clusters were combined and assessed using a gene ontology analysis to identify pathways protected through D3T pretreatment. Many of the 15 most significant pathways listed are related to neurodevelopment.

**Figure 6 antioxidants-13-00560-f006:**
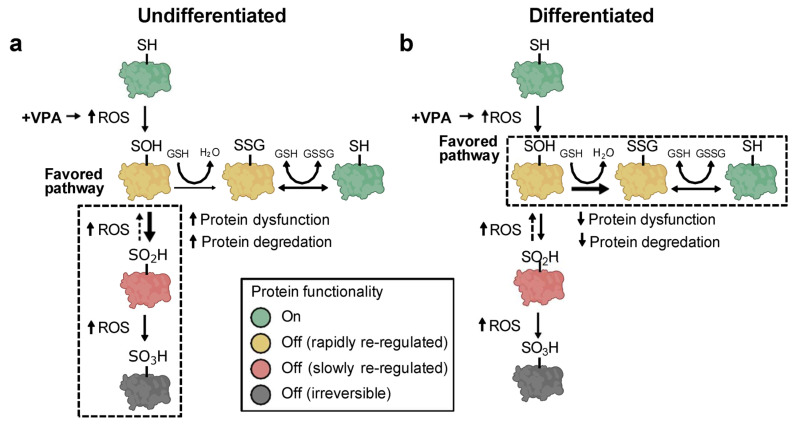
Valproic acid causes oxPTMs that are dependent upon cellular differentiation states. (**a**,**b**) Schematics of the proposed pathway by which VPA affects protein post-translational modifications in undifferentiated (**a**) and differentiated (**b**) cells. In undifferentiated cells, VPA causes extended periods of protein over-oxidation that leads to protein dysregulation and degradation. Conversely, differentiated cells are protected from severe oxidation via protein *S*-glutathionylation, enabling timely protein recovery.

## Data Availability

The sequenced samples used in this study are openly available under the NCBI BioProject ID PRJNA1093422. Lists of all differentially expressed genes presented in this manuscript are available in Appendix A. The additional data supporting the conclusions of this article will be made available by the authors on request.

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
