# Peer review of "Valproic Acid Causes Redox-Regulated Post-Translational Protein Modifications That Are Dependent upon P19 Cellular Differentiation States"

_antioxidants, 2024, doi:10.3390/antiox13050560_

Round 1
Reviewer 1 Report
The paper describes effects of VPA in a P19 cellular model of neuronal differentiation.
1. The use of the particular model is a study limitation which should be expressed in the paragraph mentioning limitations of the study.
2. The paper lacks a mechanistic idea why ROS is increased upon VPA treatment. Is this effect related to VPA or a potential metabolite, like VPA-CoA? Otherwise the scheme in Fig. 6 is highly speculative.
3. All experiments were carried out only with 5mM VPA which is a quite high concentration. A dose response investigation would greatly enhance the reliability of data.
1. Fig. 2: H2O2 also had no effect on differentiated P19-derived neurons. That requires a comment on the potential activity of defense reactions.
2. Paragraph 5.4: As mentioned above the transcriptional data require a much better presentation. Are known genes involved in redox signalling upregulated (KEAP1 etc.). To which extent NRF2 is altered?
Author Response
Re: Resubmission of manuscript ID: antioxidants-2971326
We thank you for the opportunity to submit our revised manuscript titled “Valproic acid causes redox-regulated post-translational protein modifications that are dependent upon P19 cellular differentiation states” for consideration for publication in Antioxidants.
We would also like to thank and acknowledge the reviewers for their detailed assessment of our work. We have incorporated their feedback that has significantly improved our revised manuscript.
We would first like to provide a summary of general changes made to the revised manuscript before responding point-by-point to each reviewer’s comments. The reviewers’ comments are highlighted in blue and italicized.
General manuscript changes:
- Corrected section numbering.
- Corrected formatting issues (such as spacing and incorrect text size).
- Minor rewording throughout to improve comprehensibility.
- Figure S2 was edited to add “Undifferentiated” and “Differentiated” designations above panels d and e.
- The old graphical abstract appeared incorrectly following the manuscript’s journal formatting. It has been corrected.
Reviewer #1’s comments:
- In the title and the abstract should be mentioned that a P19 cellular model of neuronal differentiation was used.
Thank you for pointing out this oversight. The title and abstract have been updated to emphasize the use of P19 cells as our model (changes are highlighted in yellow).
- VPA is a well known histone deacetylase inhibitor (https://doi.org/10.1093/emboj/cdg315). That is presumably the main reason for FVS. That needs to be mentioned in the introduction. Further, a justification would be required why redox signalling might be additionally important.
The role of VPA as a histone deacetylase inhibitor and justification for studying redox signaling during development have been addressed (page 2 lines 42–49).
- The transcriptional studies should be presented in much greater detail. That would include to provide gene lists for most up- or down-regulated genes. Are redox state relevant genes affected?
Thank you for the suggestion—we have included the full lists of up- and down-regulated genes for the presented figures as Supplementary Data 1 (attached to this revision). Redox relevant genes are largely unaffected by the various treatments (besides D3T alone, which we show induces NRF2-regulated genes in Fig. S3b). As detailed in the manuscript, we postulate that the redox regulation is occurring on a post-translational level on developmentally critical targets (such as transcription factors), and not through direct transcriptional regulation. As a result, we would not expect to see significant differential expression of redox related genes following VPA exposure.
- Minor editing of English language required.
We have edited the overall manuscript to improve comprehensibility, but we kindly ask Reviewer #1 to notify us of any specific errors that we may have missed.
- The use of the particular model is a study limitation which should be expressed in the paragraph mentioning limitations of the study.
We have added the P19 model as a limitation (page 13 lines 443–445).
- The paper lacks a mechanistic idea why ROS is increased upon VPA treatment. Is this effect related to VPA or a potential metabolite, like VPA-CoA? Otherwise the scheme in Fig. 6 is highly speculative.
We have expanded upon the epigenetic role of VPA and its ties to redox regulation and development (page 2 lines 42–49). Valproic acid promotes mitochondrial superoxide dismutase 2 acetylation, thereby decreasing the antioxidant enzyme’s activity and possibly contributing to the increased ROS observed. Regardless of the exact mechanism by which it occurs, VPA-induced oxidative stress is well documented (https://doi.org:10.1097/WNR.0000000000001663, https://doi.org:10.1080/03602530600959433, https://doi.org:10.3390/ijms241713446).
- All experiments were carried out only with 5mM VPA which is a quite high concentration. A dose response investigation would greatly enhance the reliability of data.
To determine the appropriate concentrations of both D3T and VPA to use for all experiments, we first performed single agent viability assays in both undifferentiated and differentiated cells. We decided to use 10 µM D3T for all experiments because it induces a potent NRF2 response (Fig. S3a–d) but does not affect cell viability (Fig. S2a). Similarly, we decided upon 5 mM VPA because it significantly increases reactive oxygen species generation in undifferentiated cells (Fig. S2d) but has no effect on viability (Fig. S2b). Dual treatment with both 10 µM D3T and 5 mM VPA does not affect cell viability (Fig. S2c). Furthermore, 5 mM VPA is a common concentration used in other studies (https://www.ncbi.nlm.nih.gov/pmc/articles/PMC5454960/, https://www.ncbi.nlm.nih.gov/pmc/articles/PMC8834451/, https://clinicalepigeneticsjournal.biomedcentral.com/articles/10.1186/s13148-021-01050-4, https://www.nature.com/articles/s41598-017-15165-3). We have added an explanation of the drug concentrations used to the main body of our manuscript (page 3 lines 100–102).
- Fig. 2: H2O2 also had no effect on differentiated P19-derived neurons. That requires a comment on the potential activity of defense reactions.
We addressed this astute observation (page 6 lines 236-238).
- Paragraph 5.4: As mentioned above the transcriptional data require a much better presentation. Are known genes involved in redox signalling upregulated (KEAP1 etc.). To which extent NRF2 is altered?
Please see our response to comment 3.
Reviewer #2’s comments:
- Although the mechanism of action of Valproic acid has been clarified, it remains unclear and poses various clinical problems. This research seems to shed some light on this, and it seems possible to consider not only future basic research fields but also clinical applications.
Thank you for noticing our lack of discussion on clinical applications. We have added a short section discussing the use of natural NRF2 inducers as potential therapies (page 13 lines 415–425).
- The point that requires explanation is that the concentration of the reagent used this time is one point. It is necessary to provide evidence as to whether this concentration is sufficient or not excessive.
To determine the appropriate concentrations of both D3T and VPA to use for all experiments, we first performed single agent viability assays in both undifferentiated and differentiated cells. We decided to use 10 µM D3T for all experiments because it induces a potent NRF2 response (Fig. S3a–d) but does not affect cell viability (Fig. S2a). Similarly, we decided upon 5 mM VPA because it significantly increases reactive oxygen species generation in undifferentiated cells (Fig. S2d) but has no effect on viability (Fig. S2b). Dual treatment with both 10 µM D3T and 5 mM VPA does not affect cell viability (Fig. S2c). Furthermore, 5 mM VPA is a common concentration used in other studies (https://www.ncbi.nlm.nih.gov/pmc/articles/PMC5454960/, https://www.ncbi.nlm.nih.gov/pmc/articles/PMC8834451/, https://clinicalepigeneticsjournal.biomedcentral.com/articles/10.1186/s13148-021-01050-4, https://www.nature.com/articles/s41598-017-15165-3). We have added an explanation of the drug concentrations used to the main body of our manuscript (page 3 lines 100–102).
- A minor point is the citation of literature. The format for quoting seems to be different. Better fix it.
We have tried to correct general formatting errors, but we are unsure of the specific error(s) that Reviewer #2 is referring to. If Reviewer #2 could please provide page and line numbers of the mistakes we would be happy to fix them.
Reviewer 2 Report
The paper by Piorczynski et al. seems to be a clear paper clarifying the effects of Valproic acid on the nervous system. Although the mechanism of action of Valproic acid has been clarified, it remains unclear and poses various clinical problems. This research seems to shed some light on this, and it seems possible to consider not only future basic research fields but also clinical applications. Furthermore, I have not found any major correction points from the drafting of this paper to this paper.
The point that requires explanation is that the concentration of the reagent used this time is one point. It is necessary to provide evidence as to whether this concentration is sufficient or not excessive.
A minor point is the citation of literature. The format for quoting seems to be different. Better fix it.
no
Author Response

(The authors gave the same response as above.)

Round 2
Reviewer 1 Report
All of my comments have been addressed accordingly.
See above.
Reviewer 2 Report
The authors of this paper have made appropriate corrections to the previous points. There are no further comments.
The authors of this paper have made appropriate corrections to the previous points. There are no further comments.